# Empirical Mode Decomposition as a Novel Approach to Study Heart Rate Variability in Congestive Heart Failure Assessment

**DOI:** 10.3390/e21121169

**Published:** 2019-11-29

**Authors:** Mingjing Chen, Aodi He, Kaicheng Feng, Guanzheng Liu, Qian Wang

**Affiliations:** 1Department of Biomedical Engineering, School of Basic Medical Sciences, Guangzhou Medical University, Guangzhou 511436, China; chenmj36@mail2.sysu.edu.cn; 2School of Biomedical Engineering, Sun Yat-sen University, Guangzhou 510275, China; heaod@mail2.sysu.edu.cn (A.H.); fengkch@mail2.sysu.edu.cn (K.F.); liugzh3@mail.sysu.edu.cn (G.L.)

**Keywords:** autonomic nervous system (ANS), congestive heart failure (CHF), empirical mode decomposition (EMD), heart rate variability (HRV), transfer entropy (TE)

## Abstract

Congestive heart failure (CHF) is a cardiovascular disease related to autonomic nervous system (ANS) dysfunction and fragmented patterns. There is a growing demand for assessing CHF accurately. In this work, 24-h RR interval signals (the time elapsed between two successive R waves of the QRS signal on the electrocardiogram) of 98 subjects (54 healthy and 44 CHF subjects) were analyzed. Empirical mode decomposition (EMD) was chosen to decompose RR interval signals into four intrinsic mode functions (IMFs). Then transfer entropy (TE) was employed to study the information transaction among four IMFs. Compared with the normal group, significant decrease in TE (*→1; information transferring from other IMFs to IMF1, *p* < 0.001) and TE (3→*; information transferring from IMF3 to other IMFs, *p* < 0.05) was observed. Moreover, the combination of TE (*→1), TE (3→*) and LF/HF reached the highest CHF screening accuracy (85.7%) in IBM SPSS Statistics discriminant analysis, while LF/HF only achieved 79.6%. This novel method and indices could serve as a new way to assessing CHF and studying the interaction of the physiological phenomena. Simulation examples and transfer entropy applications are provided to demonstrate the effectiveness of the proposed EMD decomposition method in assessing CHF.

## 1. Introduction

Congestive heart failure (CHF) is a common chronic cardiovascular syndrome, rendering the inability of heart to pump sufficient blood as required by the body [1]. Around 26 million people are suffering from CHF all over the world [2]. CHF may also herald a newly diagnosed cardiac disease or the deterioration of an underlying heart disease [3]. Thus the assessment for CHF patients is essential in clinical circumstance.

CHF is associated with autonomic nervous system (ANS) abnormality and fragmented patterns of the heart [4,5]. Heart rate variability (HRV), being able to access the function of ANS and some non-autonomic influence [5,6], is frequently used while investigating CHF [4]. Frequency-domain HRV indices are commonly used to analyze HRV [7,8]. Binkley et al. [9] and Pecchia et al. [6] proposed that the frequency-domain indices could indicate the autonomic imbalance characteristic of CHF. The power spectral density of HRV signal could be divided into the four bands: very low frequency VLF (0.003–0.04 Hz), low frequency LF (0.04–0.15 Hz), high frequency HF (0.15–0.4 Hz) and very high-frequency VHF (0.4–1 Hz) [10]. Power in VLF [11,12,13] has been proved to be indicators of activity of ANS, which is not susceptible to a simple autonomic branch interpretation [14]. LF band is related to the sympathetic nerve system (SNS) while the power at HF is mainly associated with the parasympathetic nerve system (PNS) [15]. Fragmented patterns, like sinus rhythm fragmentation, can contribute to those very-high frequency oscillations, and thus have an effect on cardiac contractility [5,8,16]. Though most studies mainly focused on independent changes of these indices, recent studies showed that the interactions between these bands could suggest the pathogenesis of diseases in depth. Luo et al. [17] studied directed interaction between SNS and PNS employing transfer entropy (TE), revealing the disturbed balance of the equilibrium of the ANS in CHF group. Zheng et al. [18] demonstrated that TE between LF and HF could differentiate patients with ANS dysfunction from the normal by evaluating the interaction between LF and HF.

The decomposition of HRV signals in the frequency domain is essential but hard. Linear filtering decomposition methods are traditionally utilized when analyzing HRV signals in the frequency domain [19,20]. However, cardiovascular control mechanisms are nonlinear and dynamic. Its behavior cannot be described accurately by the high-gain instability oscillation model, upon which power spectral density is predicted in linear filtering methods [21]. Besides, inaccuracy could happen because of its involvement of decomposition of low frequency component. It can lead to serious error when the frequency band is divided into several parts. Therefore, only LF and HF value are chosen in research in most cases using linear filter [17,18], while VLF and VHF values are ignored, to ensure the accuracy as much as possible. As a result, better and more refined decomposition methods are needed to obtain HRV feature accurately. Empirical mode decomposition (EMD), capable of adaptively decomposing a complex signal into a collection of relatively stationary and infinite vibration modes (intrinsic mode functions, IMFs), was proposed to deal with this problem [22,23]. Anas et al. revealed the similarity between IMFs and ECG signals and exploited EMD to separate ventricular fibrillation from other cardiac pathologies [24]. Pan et al. decomposed four subsystems from HRV signals with EMD method, of which the correspondence with the four frequency bands (VLF, LF, HF and VHF) was further validated by the Hilbert-Huang transform (HHT) [25].

Being able to estimate the causality between two variables, TE is an ideal tool to assess the flowing information among systems and subsystems [18]. Luca et al. demonstrated the ability of the TE method to gauge the interaction of simulated linear and nonlinear multivariate processes [26]. Zheng et al. inferred TE was useful to analyze respiration movement asymmetry between the left and right lungs [27]. The relationship between SNS and PNS can also be evaluated by analyzing TE between LF and HF [17,18].

As a result, the asymmetry correlation among various body functions could be better investigated utilizing the combination of EMD and TE. In this paper, the EMD method was employed to decompose HRV signals into several IMFs. Next, TE was extracted among IMFs to study the interactions between subsystems. Finally, various combinations of indices would be tested to clarify each sample as normal or CHF using Fisher’s discriminant function.

## 2. Methods

In this study, the HRV features of the normal and the CHF would be analyzed. The framework of the analysis system is shown in Figure 1. First, 24-h RR interval (the time elapsed between two successive R waves of the QRS signal on the electrocardiogram) recordings were segmented into epochs and resampled. Then, 5-min RR segments were decomposed using the EMD method. Afterwards, LF/HF ratio and TEs of the decomposed results were extracted. Finally, these indices were validated by statistical analysis and CHF assessments.

### 2.1. Data

In this study, we used electrocardiogram (ECG) recordings derived from the open database of Physionet website. The 24-h RR interval signals of 54 normal people (31 males and 23 females, aged 61.38 ± 11.63) were acquired from the Normal Sinus Rhythms RR Interval database. Forty-four CHF subjects (19 males and six females, 19 subjects’ gender were unknown, aged 55.51 ± 11.44) consisted of 15 from the Beth Israel Deaconess Medical Center (BIDMC) Congestive Heart Failure and 29 from the Congestive Heart Failure RR Interval database [28].

The first and last RR interval segments of each 24-h recordings, as well as any RR interval longer than 3 s [29], were eliminated to minimize the influence of error and noise. Considering that RR intervals of CHF patients are usually lower than normal controls [30], we do not pay particular attention to and remove the lower outliers in this study. Next, the remaining signals were divided into non-overlapping 5-min fragments. Then, each fragment was resampled to 2 Hz using the spline interpolation for applying short-term HRV analysis [4].

### 2.2. HRV Analysis

#### 2.2.1. Empirical Model Decomposition

The adaptivity and applicability of the EMD method to nonlinear and non-stationary systems has been proved for its direct extraction of energy with regard to local characteristic time scale. Four intrinsic mode functions (IMFs) would be generated from the preprocessed signals, which can yield meaningful instantaneous frequencies with Hilbert transform [22]. Each IMF should satisfy two requirements: (1) the number of extremes and zero crossings should either be the same or differ at most by one. (2) The mean value of the envelopes defined by the local maxima and by the local minima at any point is supposed to be zero [31].

Given the original signal s(t) and the algorithm is shown as follows: [32]
1.s(t) is represented as $x_{j,k}\left(t\right)\;\left(j=1,k=1\right)$ with subscript *j* represents the number of IMF and subscript k represents the time of sifting process to get ${\mathrm{IMF}}_{j}$;2.Identify all the local maxima and minima and use cubic fitting to obtain upper envelope $e_{max}\left(t\right)$ and lower envelope $e_{min}\left(t\right)$ of $x_{j,k}\left(t\right)$. Then the mean values are calculated by; (1)$$m_{j,k}\left(t\right)=\frac{e_{max}\left(t\right)+e_{min}\left(t\right)}{2}.$$3.The detail is computed by (2)$$x_{j,k+1}\left(t\right)=x_{j,k}\left(t\right)-m_{j,k}\left(t\right).$$If $x_{j,k+1}\left(t\right)$ meets the two requirements mentioned above, (3)$${\mathrm{IMF}}_{j}\left(t\right)=x_{j,k+1}\left(t\right).$$If not, repeat steps 2–3.4.To find more IMF components of input data, $x_{j+1,k}\left(t\right)\;\left(k=1\right)$ is calculated by (4)$$x_{j+1,k}\left(t\right)=x_{j,k}\left(t\right)-{\mathrm{IMF}}_{j}\left(t\right).$$If $x_{j+1,k}\left(t\right)$ is monotone, the residual signal (5)$$r\left(t\right)=x_{j+1,k}\left(t\right).$$The origin signal can be shown as (6)$$s\left(t\right)=\sum_{j=1}^{n}IMF_{j}\left(t\right)+r\left(t\right).$$If not, repeat steps 2–4.

#### 2.2.2. Indices Calculation

• LF/HF ratio:

In this study, the power spectral density of the RR intervals was computed by fast Fourier transform. The value of LF (0.04–0.15 Hz) and HF (0.15–0.4 Hz) components were chosen to calculate the LF/HF ratio [33], which is widely accepted to describe ANS balance. The formula is as follows:(7)$$\mathrm{LF}/\mathrm{HF}\;\mathrm{ratio}=\frac{\mathrm{LF}\;\mathrm{power}}{\mathrm{HF}\;\mathrm{power}}.$$

• Transfer entropy:

TE is used as an estimation technique to detect coupling strength among two time series, and the results are computed between any two in both directions. Given two sequences: [34] (8)$$\{\begin{array}{c} x^{\left(l\right)}=\left(x_{n},x_{n-u},\ldots ,\;x_{n-\left(l-1\right)u}\right)\\ y^{\left(k\right)}=\left(y_{n},y_{n-u},\ldots ,\;y_{n-\left(k-1\right)u}\right) \end{array},$$ where l and k represent the order of the Markov process, which means the future probabilities of a system are determined by its last *l* or *k* values.

The formula is proposed as:(9)$${\mathrm{TE}}_{X\to Y}=\sum p\left(y_{i+1},y_{i}^{\left(k\right)},x_{i-u-1}^{\left(l\right)}\right)log_{2}\frac{p(y_{i+1}|y_{i}^{\left(k\right)},x_{i-u-1}^{\left(l\right)})}{p(y_{i+1}|y_{i}^{\left(k\right)})},$$ where *u* is the predicting delay time. $p\left(y_{i+1},y_{i}^{\left(k\right)},x_{i-u-1}^{\left(l\right)}\right)$ represents the joint probability of $y_{i+1},y_{i}^{\left(k\right)},x_{i-u-1}^{\left(l\right)}$ while $p(y_{n+1}|y_{i}^{\left(k\right)},x_{i-u-1}^{\left(l\right)})$ and $p(y_{n+1}|y_{i}^{\left(k\right)})$ are the conditional probabilities. Considering the Markov properties of our data, k and l were defined as 1 while u equaled 0 in this study. Then, the equation can be expressed as follows: (10)$${\mathrm{TE}}_{X\to Y}=\sum p\left(y_{i+1},y_{i},x_{i}\right)log_{2}\frac{p(y_{i+1}|y_{i},x_{i})}{p(y_{i+1}|y_{i})}.$$

#### 2.2.3. Simulated Data of EMD

Based on the local characteristic time scale of the data, EMD is an adaptive and efficient decomposing method on extraction of features for nonlinear and nonstationary signals [23]. In order to validate the superiority of the EMD method in the face of various oscillatory signals, the performance of EMD and the linear filtering method was compared.

In the simulated experiment, we compared the performance of the linear filter and EMD facing nonlinear signal. The simulated experiment was conducted on autoregressive (AR) processes as the following system [35]: (11)$$LF_{n}=\sum_{i=1}^{k}\alpha_{i}LF_{n-i}+J\vartheta_{n}.$$
(12)$$HF_{n}=\sum_{i=1}^{l}\beta_{i}HF_{n-i}+K\varepsilon_{n}+\gamma \sum_{i=1}^{l}\delta_{i}LF_{n-i}.$$

In order to simulate the ANS signals, the frequency band of LF oscillations was set with $f_{LF}\in$ [0.04, 0.15] while HF oscillations with $f_{HF}\in$ [0.15,0.4]. $\vartheta_{n}$ and $\varepsilon_{n}$ stand for Gaussian white noises with mean of zero and variance of 1. $\alpha_{i}$, $\beta_{i}$ and $\delta_{i}$, calculated based on the Akaike information criterion (AIC), are the main parameters of this autoregressive model. J and K was set to 0.1 and k = *l* = 10 in this model. γ, which was set to 0.5, was caused by the time delay of the LF sequence. Characteristics of this synthesized signal in time and frequency domain are shown in Figure 2A,B. The frequency spectrum of LF and HF component are drawn in Figure 2C,D for ease of comparison.

### 2.3. Statistical Analysis

Statistical analysis was performed to verify whether the features showed significant difference between different groups via one-way ANOVA [36]. Further, Fisher’s discriminant function would classify each sample as normal or CHF. The significance test and discriminant analysis test were performed using SPSS version 22.0.0.0 (SPSS Inc., Chicago, IL, USA).

## 3. Results

### 3.1. Simulated Analysis by EMD

From the simulated results, EMD was superior to linear filtering method when facing the simulated ANS signals. The results are shown in Figure 2E,H. From this perspective, EMD had especially better performance than linear filter when it comes to low frequency components. The EMD spectrum, though still had slight difference with the original signal, conformed more to the raw components. Nevertheless, the linear filter brought a particularly inaccurate outcome in the low frequency band, which is shown in Figure 2J. The energy leakage of linear filter may be a reason for this large error. As for the HF components, both linear filter and EMD had good performance and decomposed the synthesized signal successfully. Generally speaking, the linear filtering method presented more serious error in this experiment, especially in low frequency band. EMD was preferable to linear filter processing simulated nonlinear signals. In the above simulation, it has been proved that EMD is a suitable way to investigate HRV signals.

### 3.2. HRV Analysis by EMD

The results of EMD of a typical 5-min HRV segment are shown in Figure 3. The first four IMFs contained almost the entire energy of the original signal. Hence, the first four IMFs of the HRV signal of each subject were selected for further analysis. 

TEs among all IMFs were calculated and TE (X→Y) estimates information transfer from processes X (driver system) to Y (target system) [18]. The mean ± standard deviation (SD) values of TE are listed in Table 1. Averages of all indices of CHF group were lower compared with the normal. The TE (2→1), TE (3→1), TE (4→1) and TE (3→4) showed a very significant difference (*p* < 0.001), and TE (3→2) showed a significant difference (*p* < 0.05) between the normal and CHF groups.

Figure 4 shows the results of TE (*→1; the target system of TEs are IMF1) between the normal and CHF groups. The information flew from any other IMFs to IMF1 of normal people was obviously more than CHF groups. The value of TE (3→*; the driver system of TEs are IMF3) is shown in Figure 5. Statistically, these indices showed the information transferring from IMF3 to other IMFs of normal group was obviously more than CHF subjects.

### 3.3. Disease Analysis by TE

TE (2→1), TE (3→1), TE (4→1), TE (3→4) and TE (3→2) were chosen as representative indices in CHF assessment according to above results, along with the LF/HF ratio, which has already been proved a powerful index in the past study [6]. The Fisher’s discriminant function was carried out to classify all the 98 samples. The accuracy (Acc), sensitivity (Sen), specificity (Spe) are defined as the percentage of correctly classified samples, correctly classified CHF samples and correctly classified healthy samples respectively. As demonstrated in Table 2, the combination of TEs the target systems of which are IMF1 (TE (2→1), TE (3→1), TE (4→1)) and of TEs the driver systems of which are IMF3 (TE (3→1), TE (3→2), TE (3→4)) both achieved a better performance than discrimination based on separate features. The combination of all the six indices reached the best performance with 85.7% Acc and 85.2% Spe. Furthermore, as the receiver operating character (ROC) curves shown in Figure 6, the combination of all indices led to the highest area under the curve (AUC) value (0.8570).

As demonstrated in the results, the combination of LF/HF, TE (2→1), TE (3→1), TE (4→1), TE (3→2) and TE (3→4) can classify the CHF from the normal with the highest accuracy in this study.

## 4. Discussion

CHF is a chronic cardiovascular syndrome related to dysfunction of ANS and erratic rhythm [4,5,17]. Linear methods are usually applied to analyze HRV signals [19,20,21] and detect CHF [18] in the past. However, a consensus seems to emerge in recent years that autonomic nerve system is non-linear and dynamic [37,38]. As a result, new applicable methods are needed to analyze such signals. In the present study, we explored the most appropriate method to analyze HRV signals in the CHF group.

### 4.1. Motivation of EMD

EMD is a reliable method for nonlinear and non-stationary signals like ANS signals [22,23,39,40]. In our study, the superiority of EMD was proved in the simulation. Unlike traditional linear filtering method, which would lead to inaccurate decomposition of LF components and cause an especially large error in boundary area, EMD showed a good performance dealing with the simulated ANS signals. EMD was employed in previous studies due to its efficiency and accuracy. Shafqat et al. [41] reported EMD was a potential tool to analyze physiological signals. Acharya et al. [37] used EMD to improve accuracy of detection of people with CHF alarming. Thus, EMD is a suitable way to investigate HRV signals and to differentiate CHF patients from normal group.

### 4.2. Analysis of IMFs

Any complicated data set can be decomposed into finite components (IMFs) through EMD, which contain the local characteristic signals of the original signal at different time scales [38]. The IMF components are able to represent the energy in specific frequency bands when the central frequency lies within the band limits and the standard deviations are less than twenty percent outside the boundary [41]. By the Hilbert transform (HT), instantaneous attributes of different time series were calculated [25,42]. The instantaneous frequency is no longer affected by the unnecessary fluctuation caused by the asymmetric waveform thanks to the modified requirements, which makes EMD more suitable for nonstationary signals [37]. Furthermore, Pan et al. extracted instantaneous frequency of IMFs and demonstrated their correspondence with four physiological subsystems through HT [25]. The results explained why IMFs were meaningful and could stand for the energy in specific frequency bands. We obtained the first four IMFs through EMD method. By HT, the instantaneous frequency (IF) parameters of IMFs were obtained. The mean value (MIF) and standard variation of IFs are shown in Table 3. IMF1 corresponded to the VHF component. Physiological significance of VHF band is still controversial, while it is reported recently that VHF could represent erratic rhythm of heart [5,16]. Wang et al. also reported that VLF was associated with heart contractility in the research [8]. IMF2 was the HF component and IMF3 was the LF component, which are influenced by PNS and SNS respectively IMF4 was the VLF component. Constituting the majority of total power in HRV, VLF band is a powerful risk predictor of CHF [11]. It has been reported that power in VLF band can reflect the activity of ANS [11,12,13]. VLF power is speculatively related to thermoregulatory mechanisms, the renin–angiotensin system and peripheral chemoreceptors in the past studies [29,43,44] Therefore, the first four IMFs corresponded to four frequency components of the HRV signal respectively.

### 4.3. Physiological Significance

CHF disease is a chronic cardiovascular syndrome along with ANS dysfunction and erratic patterns [4,16]. There are usually more complex respiratory pattern [25], imbalance of ANS [18], sinus rhythm fragmentation [5] and morpho-functional disorders [45] in people with CHF. HRV has been demonstrated to be a noninvasive tool to estimate the function of ANS and erratic rhythm of the heart [5,6,16]. Previous studies have investigated the power changes in HRV, revealing the tone of HRV is different between CHF patients and normal controls [6,9,17].

Transfer entropy can estimate the amount of uncertainty reduced in future values of one time series by knowing the past values of another series and thus quantify different coupling zones of physiological systems [17,27]. Since IMFs can represent different physiological subsystems to some degree, TE values among IMFs can calculate information transaction among these subsystems, and thus estimate the coupling strength. As a result, TE could stand as an important marker of modulation in this study. TEs between any two IMFs in CHF patients are lower in value than normal group, which suggested that the interaction of CHF subjects, not only the intra autonomic system but also between ANS and non-autonomic components, is weakened. Vincenza et al. reported the absence of a significant influence of feedback on the heart in cardiac patients [46]. Furthermore, TE (*→1) and TE (3→*) shows a significant difference between CHF subjects and normal ones. The estimation of TE (*→1) is significantly lower in CHF subjects, revealing less amount of information transmission to the VHF band in CHF patients. Those very-high frequency oscillations are usually referred to as fragmented patterns of heart, the apparent dynamical signature of which is frequent changes in heart rate acceleration sign in CHF group [5]. This weak coupling strength between ANS (represented by HF, LF and VLF) and non-autonomic components (represented by VHF) suggests a physiological disorder happening in CHF group. It can be inferred that the control of ANS over non-autonomic components is weakened after perturbation, resulting in an increase in the overall amount of short-term variability [5]. TE (3→*) in the CHF group is also lower than the normal. The less information flowing from LF to HF and VLF implies the reduced transmission among ANS branches and thus an ANS dysfunction, consistent with past studies [26,47]. Besides, Liu et al. [36] and Rovere et al. [48] reported that LF proportion was reduced in CHF subjects. In conclusion, as an innovative indicator, TE provides a new insight into the complicated interaction among physiological systems.

### 4.4. Limitation

This study had some limitations. First, although the combination of indices could differentiate CHF subjects from normal people, the severity of multistage risk was not clarified. Second, though LF component has been commonly seen as the reflection of the combination of PNS and SNS activity [49], it only represented SNS activity in our study in order to simplify the analysis process. Finally, age and gender were not consistent in CHF and control groups. Since it was not significantly different, it might still have some influence on the results. Thus, this limitation would be taken into consideration in future study.

## 5. Conclusions

In this study, EMD was applied to decomposing RR intervals of the CHF group. The TE method was used afterwards to investigate the interaction features. This paper defined EMD as an effective tool to analyze RR intervals signals and TE a powerful marker to detect physiological modulation. Moreover, the results suggested the weakened coupling zones not only intra autonomic system but also between ANS and non-autonomic components, inferring a physiological disorder in CHF patients. Therefore, this study provided a new sight into assessing CHF and interaction of physiological phenomena.

## Figures and Tables

**Figure 1 entropy-21-01169-f001:**
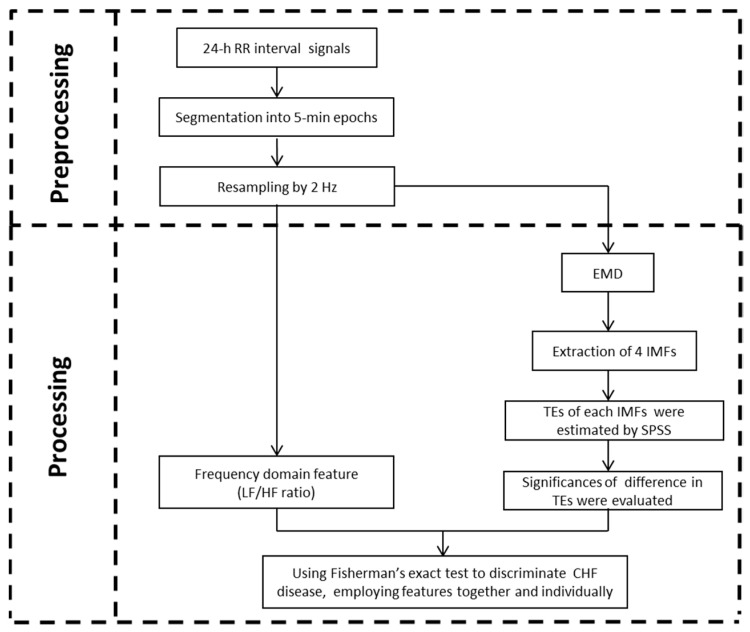
The framework of the proposed signal processing.

**Figure 2 entropy-21-01169-f002:**
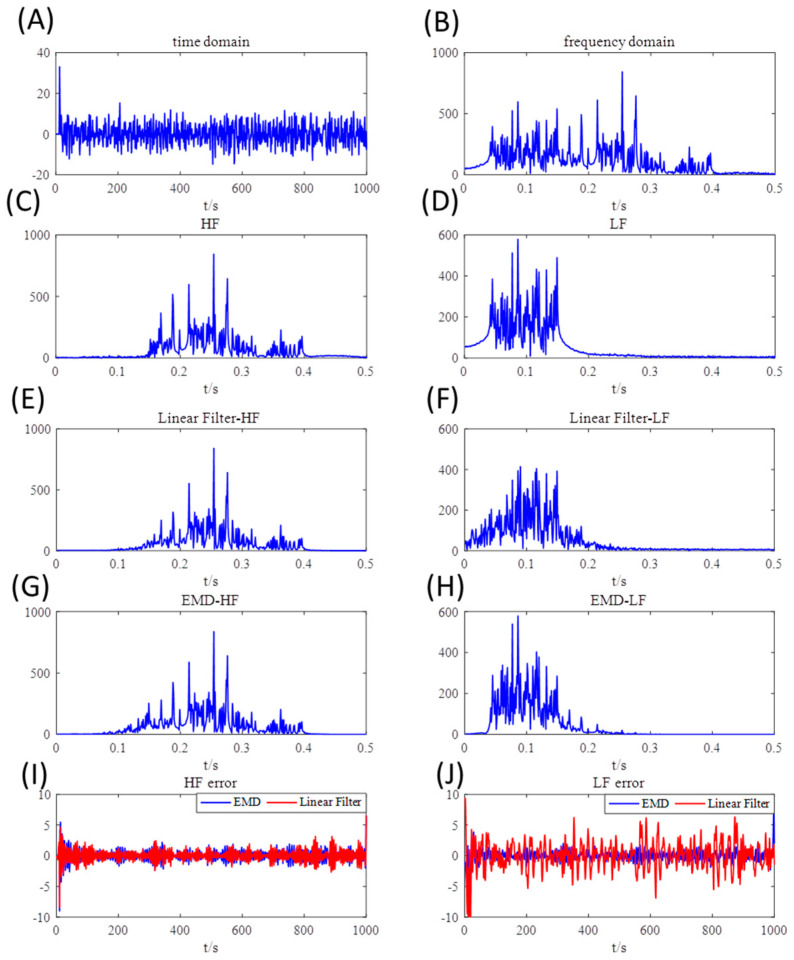
Decomposition results of the simulated heart rate variability (HRV) signals. (**A**,**B**): The synthesized signal in the time domain and frequency domain. (**C**,**D**) The high frequency (0.15–0.4 Hz) and low frequency (0.04–0.15 Hz) components in frequency domain. (**E**,**F**): Decomposition results of linear filter. (**G**,**H**): Decomposition results of empirical mode decomposition (EMD). (**I**,**J**): Error between the original component signals and decomposed signals in the high and low frequency band.

**Figure 3 entropy-21-01169-f003:**
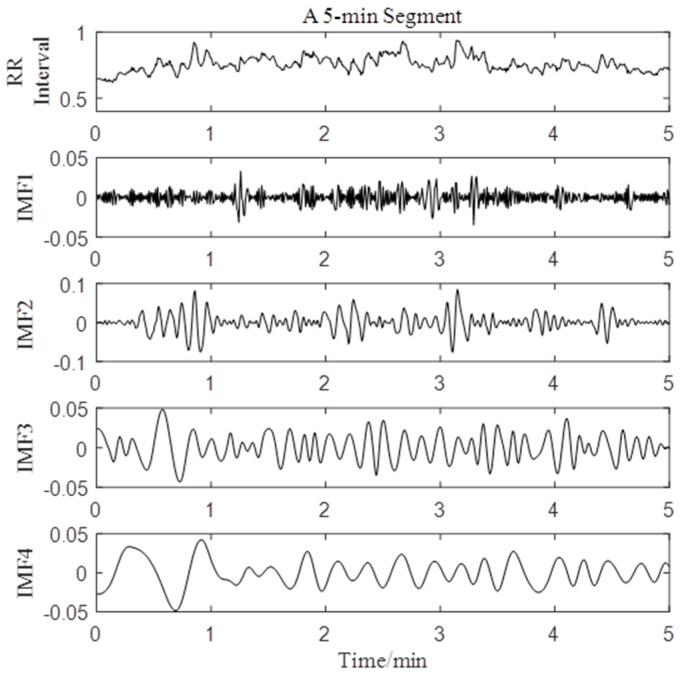
The results of EMD of a typical 5-min HRV segment.

**Figure 4 entropy-21-01169-f004:**
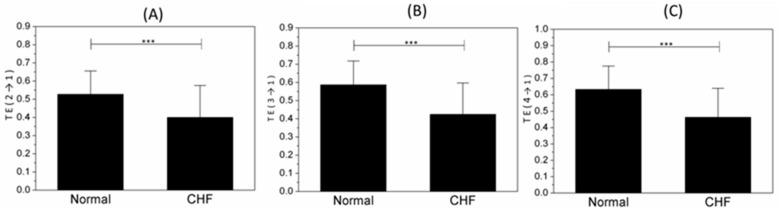
Indices for normal and congestive heart failure (CHF) groups. (**A**) TE (2→1), (**B**) TE (3→1) and (**C**) TE (4→1). *** represents *p* < 0.001.

**Figure 5 entropy-21-01169-f005:**
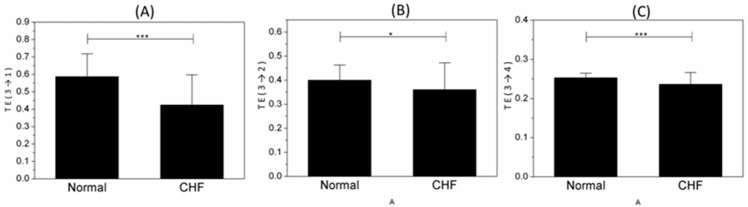
Indices for normal and CHF groups. (**A**) TE (3→1), (**B**) TE (3→2) and (**C**) TE (3→4). *,*** represent *p* < 0.05, *p* < 0.001, respectively.

**Figure 6 entropy-21-01169-f006:**
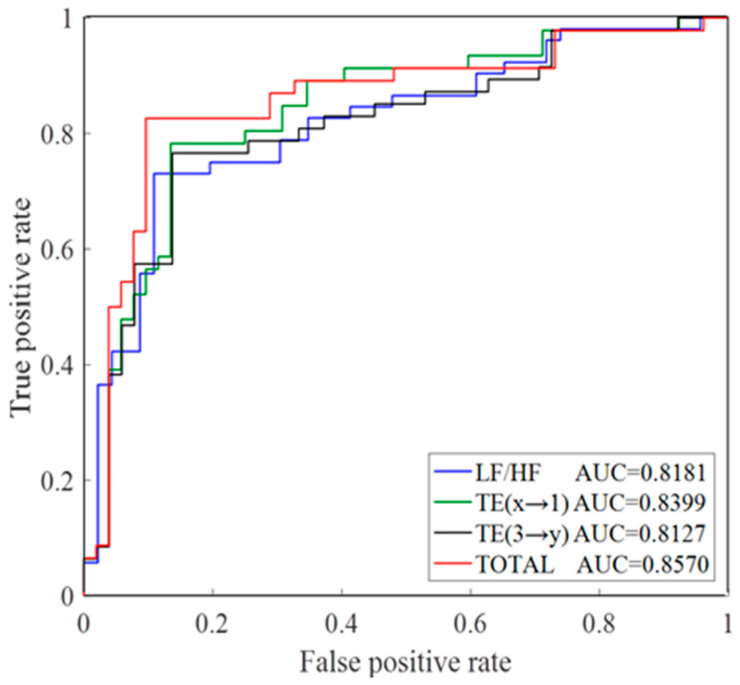
The receiver operating character (ROC) curves. TE (x→1) means the combination of TE (2→1), TE (3→1) and TE (4→1). TE (3→y) means the combination of TE (3→1), TE (3→2) and TE (3→4). TOTAL means the combination of LF/HF, TE (2→1), TE (3→1), TE (4→1), TE (3→2) and TE (3→4).

**Table 1 entropy-21-01169-t001:** Values of transfer entropy (TE) among intrinsic mode function (IMF)1, IMF2, IMF3 and IMF4.

Indices	Normal (Mean ± SD)	CHF (Mean ± SD)	*p*-Value
TE (2→1)	0.5275 ± 0.12805	0.3995 ± 0.17607	0.000 ***
TE (1→2)	0.3523 ± 0.08306	0.3313 ± 0.13262	0.364
TE (3→1)	0.5866 ± 0.13141	0.4245 ± 0.17261	0.000 ***
TE (1→3)	0.2664 ± 0.03964	0.2531 ± 0.07634	0.299
TE (4→1)	0.6323 ± 0.14308	0.4625 ± 0.17730	0.000 ***
TE (1→4)	0.2161 ± 0.02868	0.2010 ± 0.04566	0.060
TE (3→2)	0.3999 ± 0.06272	0.3600 ± 0.11196	0.039 *
TE (2→3)	0.2770 ± 0.03107	0.2623 ± 0.06932	0.198
TE (4→2)	0.4283 ± 0.06653	0.3972 ± 0.10781	0.099
TE (2→4)	0.2205 ± 0.01821	0.2102 ± 0.03837	0.109
TE (4→3)	0.3442 ± 0.02340	0.3269 ± 0.05502	0.056
TE (3→4)	0.2526 ± 0.01196	0.2359 ± 0.03023	0.001 ***

*, *** represent *p* < 0.05 *p* < 0.001 between normal and CHF groups, respectively.

**Table 2 entropy-21-01169-t002:** Performance of classification.

Indices	Acc (%)	Sen (%)	Spe (%)
LF/HF	79.6	86.4	74.1
TE (2→1)	69.4	70.5	68.5
TE (3→1)	70.4	70.5	70.4
TE (4→1)	70.4	70.5	70.4
TE (3→2)	63.3	63.6	63.0
TE (3→4)	78.6	75.0	81.5
TE (2→1) and TE (3→1) and TE (4→1)	81.6	81.8	81.5
TE (3→1) and TE (3→2) and TE (3→4)	80.6	81.8	79.6
LF/HF and TE (2→1) and TE (3→1) and TE (4→1) and TE (3→2) and TE (3→4)	85.7	86.4	85.2

**Table 3 entropy-21-01169-t003:** Frequency parameters.

Symbol	MIF ± SD (Hz)	Frequency Band
IMF1	0.431 **±** 0.152	VHF
IMF2	0.199 **±** 0.081	HF
IMF3	0.076 **±** 0.037	LF
IMF4	0.027 **±** 0.012	VLF

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
