# Peer review of "Empirical Mode Decomposition as a Novel Approach to Study Heart Rate Variability in Congestive Heart Failure Assessment"

_entropy, 2019, doi:10.3390/e21121169_

Round 1

Reviewer 1 Report

The article is original, interesting but need some minor corrections, by a native English language speaker. Eg. row 33 - 26 billion (exceed 5 times World population) should be millions.

row 61 - HRV is finer?- find adequate expression

Reviewer 2 Report

The idea of calculating TE from IMFs is interesting. However, the paper should be carefully revised by the authors to clarify some methodological aspects and to correct some physiological interpretations.

Specific Comments

Section 1

Line 44: the authors say that very high-frequency (VHF) components of HRV are influenced by the autonomic system. Further on, the authors say that ‘cardiac contractility’ contributes to VHF rhythms. This is not much clear, and I would expect only non-autonomic influences in this band, when it is present. Could you please clarify it?

Section 2

Line 85 and Figure 1: why did the authors resample the data to 2 Hz? Is this a requirement to EMD or only to Fourier transform? In line 104, the authors say IMFs were generated from original signals.

Section 2.1

Line 97: did the authors check for the presence of outliers after the preprocessing? Removing intervals longer than 3 s may not exclude all artifacts. For example, premature contractions generate artefactual intervals with values lower than the normal.

Section 2.2.1

The algorithm for EMD is somewhat confusing, please, revise it. Some questions:

In the first step (k=1), x_{j,k}(t) = s(t)? Over what signal is m(t) calculated? x_{j,k}(t)? In step 4, what do you mean by ‘the origin signal is redefined’?

Section 2.2.2

Definition of TE:

What is ‘l’ and ‘k’ in Eq. 8? If they are the length of x and y, they cannot be set to 1. It does not make sense. If u = 0 (as assumed by the authors), x and y are series of constant values (x_n and y_n repeatedly).

Section 2.2.3

I could not understand what Eqs. 11 and 12 are. What do LF and HF represent in the equations? What is the meaning of ‘n’ in these equations? How did you map LF and HF oscillations to the corresponding IMFs? How did you generate the signal in Fig 2A? How did you calculate the spectrum in Fig 2B? Considering this spectrum was calculated using Fourier transform, you are using a linear decomposition as the gold standard, which does not make much sense to compare with the other algorithms.

Section 3.3

How did you calculate the ROC curves when several indices are combined?

Section 4

Line 227: automatic nerve system à autonomic nervous system Table 3: as the authors previously stated, VLF band may be influenced by several mechanisms, not only the peripheral nervous system. Again, what do you mean by cardiac contractility influencing HRV at VHF band? Consider looking at (Front Physiol 8:255, 2017) for a reasonable interpretation of ultra-rapid oscillations in HRV. Line 261: in general, the predictability of HRV in CHF is lower than in healthy subjects (see Phys Rev E Stat Nonlin Soft Matter Phys 71:021906, 2005). Therefore, I do not believe that increased unpredictability is the reason for the diminished coupling between IMFs. The discussion section should be carefully revised, regarding the physiological interpretation. It is based on the factors presented in Table 3 and this table may not be fully correct. Line 281: PNS à SNS

Section 5

Why would TE represent a marker of autonomic modulation? It should be understood from its definition: information transfer from one signal to another.

Round 2

Reviewer 2 Report

None

Author Response

Thank you for your comments! We are really sorry for we don't address your concerns about physiological interpretation of some results last time. A point-by-point response is in the attachment.

Looking forward to your reply!

Round 3

Reviewer 2 Report

The authors seem to have addressed the issues with more attention now.

To me, the authors still should do the following:

1) Their interpretation of VHF are still not correct. I agree that the physiological meaning of VHF is controversial, but I do not agree with their statement that "consensus emerge that VHF is a robust sign of ANS dysfunction". The papers I have recommended to them (see, e.g. [Front Physiol 8:255, 2017], [J Cardiovasc Electrophysiol, Vol. 16, pp. 954-959, September 2005]) point clearly to the interpretation of VHF as non-autonomic influences. Those very-high frequency oscillations are usually referred to as erratic or fragmented patterns. This must be taken into account in their discussion to keep readers up to date and not to falling into sterile discussions.

2) The explanation to point 3 given in their last response is nice and should be added to the paper.

If the authors agree to it, I can give my green light for publication.

Author Response

We really appreciate your comments and have benefited a lot! The physiological interpretation of VHF and the discussion part are carefully revised in the manuscript. The point-to-point response is in the attachment.
